# Increasing donor-acceptor spacing for reduced voltage loss in organic solar cells

Jing Wang [1], Xudong Jiang[2,3], Hongbo Wu[1], Guitao Feng[2,3], Hanyu Wu[1], Junyu Li[4], Yuanping Yi [3], Xunda Feng [1], Zaifei Ma[1], Weiwei Li [2,3✉], Koen Vandewal [5✉] & Zheng Tang [1✉]

The high voltage losses ($V_{loss}$), originating from inevitable electron-phonon coupling in organic materials, limit the power conversion efficiency of organic solar cells to lower values than that of inorganic or perovskite solar cells. In this work, we demonstrate that this $V_{loss}$ can in fact be suppressed by controlling the spacing between the donor (D) and the acceptor (A) materials (DA spacing). We show that in typical organic solar cells, the DA spacing is generally too small, being the origin of the too-fast non-radiative decay of charge carriers ($k_{nr}$), and it can be increased by engineering the non-conjugated groups, i.e., alkyl chain spacers in single component DA systems and side chains in high-efficiency bulk-hetero-junction systems. Increasing DA spacing allows us to realize significantly reduced $k_{nr}$ and improved device voltage. This points out a new research direction for breaking the performance bottleneck of organic solar cells.

[1] State Key Laboratory for Modification of Chemical Fibers and Polymer Materials, Center for Advanced Low-dimension Materials, College of Materials Science and Engineering, Donghua University, Shanghai 201620, P. R. China. [2] Beijing Advanced Innovation Center for Soft Matter Science and Engineering & State Key Laboratory of Organic-Inorganic Composites, Beijing University of Chemical Technology, Beijing 100029, P. R. China. [3] Key Laboratory of Organic Solids, Institute of Chemistry, Chinese Academy of Sciences, Beijing 100190, P.R. China. [4] DSM DMSC R&D Solutions, P.O. Box 18, 6160 MD Geleen, The Netherlands. [5] Instituut voor Materiaalonderzoek (IMO-IMOMEC), Hasselt University, Wetenschapspark 1, BE-3590 Diepenbeek, Belgium. ✉email: liweiwei@iccas.ac.cn; koen.vandewal@uhasselt.be; ztang@dhu.edu.cn

The performance of donor-acceptor (DA) organic solar cells based on the bulk-heterojunction (BHJ) concept[1,2] has been improving rapidly[3–10], as a result of the development of high-performance donor and acceptor materials with better optical, electrical, and morphological properties[5,7,8,11–14]. Peak external quantum efficiencies (EQEs) and fill-factor (FF) have now exceeded 80% for organic solar cells[10,15–17]. Nevertheless, the open-circuit voltage ($V_{OC}$) is still much lower than that predicted by Shockley–Queisser theory[18,19]. The low $V_{OC}$, or in other words, the high voltage loss ($V_{loss}$) limits the power conversion efficiency (PCE) of organic solar cells[20–24].

$V_{loss}$ consists of radiative ($\triangle V_r$) and non-radiative recombination voltage loss ($\triangle V_{nr}$), which for the DA organic solar cells are related to the properties of the charge transfer (CT) states formed at the DA interfaces[25–28]: While the radiative decay rate of the CT states ($k_r$) determines $\triangle V_r$[22,29,30], the ratio of $k_r$ and the non-radiative decay rate of CT states ($k_{nr}$) determines $\triangle V_{nr}$[22,27,31–35].

The main reason for the high $V_{loss}$ in the organic solar cells is the high $\triangle V_{nr}$[21–24,27,32,36–40], and reducing it has been proven to be highly challenging. An effective strategy to reduce $\triangle V_{nr}$ is to reduce the energetic difference between the local excited states ($S_1$) in the pristine donor or acceptor material and CT states, resulting in hybridization and an increase in the transition oscillator strengths of the $S_1$-CT hybrid states[35,41–45]. However, the reduction in $\triangle V_{nr}$ is in that case mostly the result of a higher $k_r$, leading to increased $\triangle V_r$[35,44]. Thus, $V_{loss}$ does not necessarily reduce, even though a reduced $\triangle V_{nr}$ is achieved using this method.

The key to reduce $\triangle V_{nr}$, and realize a reduced $V_{loss}$, is to eliminate the non-radiative decay paths and reduce $k_{nr}$ without increasing $k_r$[22,23,32,34]. This requires an understanding of the origin of the generally high $k_{nr}$ in DA organic solar cells. Recently, it was suggested that the high $k_{nr}$ in DA blends is related to the presence of the high-frequency carbon-carbon vibrational modes, leading to a strong vibrational coupling between CT and ground states[22]. Later, it was demonstrated that $k_{nr}$ and $V_{loss}$ were reduced in blends based on organic visible light-emitting materials, due to their higher energy of CT state ($E_{CT}$)[38], compared to that of the solar cell materials, reducing the degree of vibrational wavefunction overlap, and thus the degree of coupling between the vibrational states. However, for solar light harvesting, it is highly undesired to realize a reduced $k_{nr}$ by increasing $E_{CT}$, since the use of high gap organic semiconductors (required for high $E_{CT}$) limits the spectral response range, and thus the photocurrent under solar illumination.

The classic theory for the non-adiabatic electronic transitions, as is used to describe non-radiative decay, starts with

$$k_{nr} = \frac{4\pi^2}{h} V^2 FC \qquad (1)$$

where $h$ is the Planck constant, $V$ is the electronic coupling matrix element, and FC is the Frank–Condon factor describing the vibrational overlap. Including the high-frequency vibrational modes leads to increased FC, being the main reason for the high $k_{nr}$ in organic photovoltaics. Although, $k_r$ also depends on FC and the transition dipole moment ($M$)[24], it is possible to reduce $k_{nr}$ without increasing $k_r$. This requires reducing $V$, a parameter exponentially dependent on the spacing between the donor and the acceptor molecules (DA spacing) forming the CT states[46,47]. A small DA spacing leads to a high $V$ value, and thus a high $k_{nr}$[48,49]. On the other hand, a too large DA spacing will lead to inefficient dissociation of the $S_1$ states, causing geminate recombination of excitons and limiting the yield of free charge carriers[41,50–54]. Currently, most of the high-efficiency organic solar cells are with high peak EQEs, which suggests that the value of DA spacing does not exceed that required for efficient electron transfer from D to A. However, it is unclear whether the DA spacing in organic blends is too small with respect to the maximum spacing tolerated, maintaining efficient charge transfer, and if it can be increased in favor of a reduced $k_{nr}$.

To answer the above question, the relationship between DA spacing and the rates for CT state formation and recombination needs to be established. In organic light-emitting diodes based on the thermally activated delayed fluorescence (TADF) exciplex emitters, tuning of the DA spacing has been achieved via engineering the chemical structure of the active material or the microstructure of the active layer[55]: Increasing the DA spacing was found to give rise to the reduced energy difference between the triplet and singlet CT states in the DA blend, and thus, increased device emission efficiency. However, for organic solar cells, the investigation regarding the DA spacing has been arduous, due to a lack of methods to reliably identify and tune the DA spacing in BHJ organic blends. In this work, we achieve a variation of the DA spacing by molecular structural engineering of the active materials, and we identify that the high $k_{nr}$ in organic solar cells is indeed associated with a too small DA spacing. This is first demonstrated in solar cells based on double-cable donor-acceptor (DCDA) polymers, in which the donor and the acceptor molecules are chemically linked by alkyl chain linkers: A correlation between the length of the linkers and the DA spacing is observed, and both $k_{nr}$ and $V_{loss}$ of the solar cell reduce with increasing DA spacing. For solar cells based on the more conventional BHJ systems, increased DA spacing is realized by modifying the side chains of the donor and the acceptor materials: For DA spacings up to an estimated 5 Å, $k_{nr}$ is also found to reduce with increasing DA spacing, leading to a reduction of $V_{loss}$ by as much as 0.2 V. Since the dissociation efficiency of $S_1$ states in the blends with increased DA spacing remains high, we conclude that the DA spacing must be generally too small in the solar cells studied in this work. As the BHJ systems studied are used in the current state-of-the-art organic solar cells, this newly discovered origin for the high $k_{nr}$ is likely also present in the currently highest efficiency devices, and overcoming it by increasing the DA spacing is expected to be a promising strategy to push PCEs over the benchmark value of 20%.

## Results

**Structural analysis for the thin films of DCDA polymers.** In order to reliably tune the DA spacing, we first utilize DCDA polymers for which the donor backbone is covalently linked to acceptor units using non-conjugated linker groups[56–58]. The DCDA polymers are a good model system for investigating the impact of DA spacing on CT state properties and $V_{loss}$ in organic solar cells, since as compared to BHJ systems, they have relatively more stable and predictable morphological and crystalline properties, due to the spatial confinement of the donor and the acceptor molecules by chemical bonding[58–61]. The DCDA polymers studied in this work are based on a donor component similar to PBDB-T (poly[[2,6-(4,8-bis(5-(2-ethylhexyl)thiophen-2-yl)-benzo[1,2-b:4,5-b′]dithiophene))-alt-(5,5-(1′,3′-di-2-thie-nyl-5′,7′-bis(2-ethylhexyl)benzo [1′,2′-c:4′,5′-c′]dithiophene-4,8-dione)]])[3], linked to the acceptor component NDI (naphthalene diimide) by alkyl chain linkers based on different numbers of methylene groups (Fig. 1a). Both the donor and acceptor units in the DCDA polymers are commonly used as building blocks for efficient BHJ organic solar cells[3,4,62,63], and their synthetic routes are provided in Supplementary Note 1.

To evaluate the nano-scale packing structures in the thin films of the DCDA polymers, transmission electron microscope (TEM) measurements are performed. Prior to the measurements, the thin films were stained by the vapor of a 0.5 wt% $RuO_4$ solution to enhance imaging contrast. Due to the selective staining of

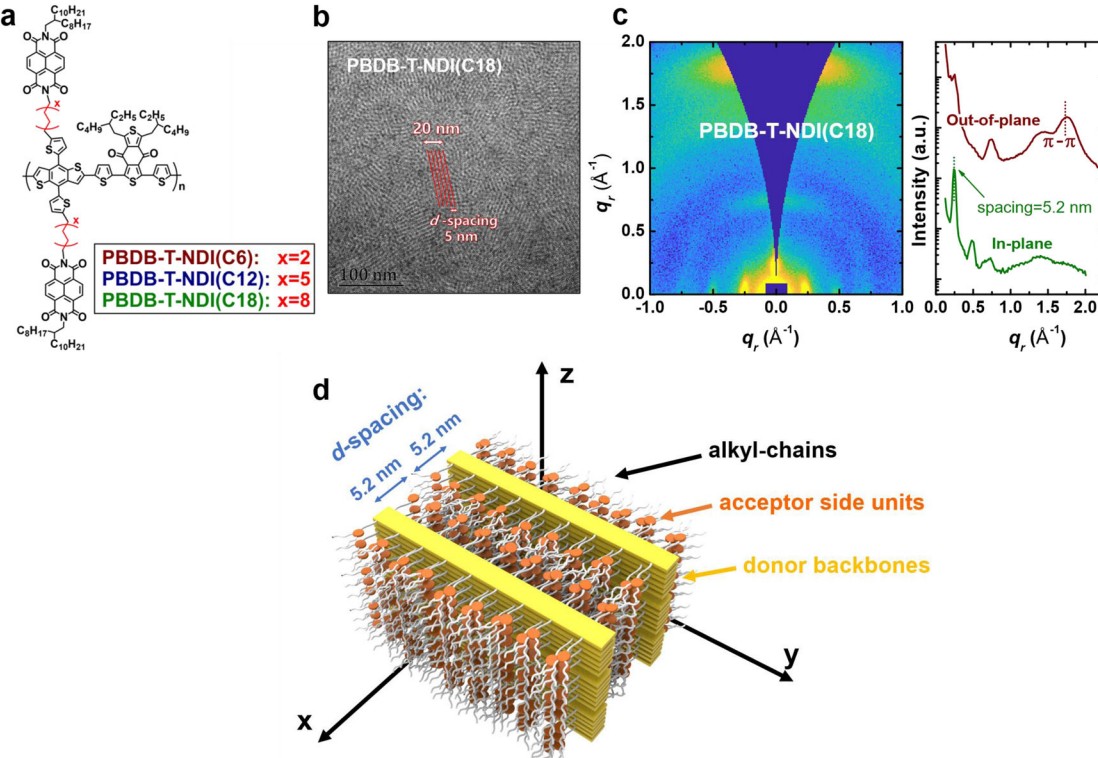

**Fig. 1 Structural analysis for the thin films of DCDA polymers. a** Chemical structure of the DCDA polymers with different lengths of alkyl chain linkers. **b** TEM image and **c** GIWAXS pattern from the thin film based on PBDB-T-NDI(C18). **d** Schematic representation of the molecular arrangement in the ordered phases of the thin film based on PBDB-T-NDI(C18).

aromatic species by $RuO_4$, the backbone of the polymer fully consisting of aromatic rings appears to be darker than the side units, as shown in Fig. 1b, as well as in Supplementary Fig. 10: Periodic patterns of black and white stripes can be observed in the TEM images, indicative of a high degree of structural order. The ordered stripes correspond to vertically oriented lamellar assemblies of the polymer with a $d$-spacing of ≈5 nm, where the backbones and the side units segregate alternately. The vertical orientation of the lamellar structures is also confirmed by grazing incident wide-angle diffraction (GIWAXS) patterns showing equatorial scattering at $q_x = 0.25\,\text{Å}^{-1}$, being the second-order diffraction from the lamellae. Furthermore, in-plane stacking of the aromatic backbones is observed, as shown by the strong meridional scattering at $q_z = 1.75\,\text{Å}^{-1}$ corresponding to a spacing of 3.6 Å (π-π). Moreover, the GIWAXS pattern shows a vertical streak located at $q_z = 0.25\,\text{Å}^{-1}$. This scattering suggests structural order along the film normal, i.e., the direction of the π-π stacking, with a spacing of 2.6 nm approximating to the parallel stacking of seven backbones. The segmentation of the aromatic stacking originates from the incommensurability of the side units with alkyl chains and the rigid backbones with aromatic rings, a behavior commonly observed in supramolecular systems such as discotic liquid crystals[64].

The assembled structure of the DCDA polymer (PBDB-T-NDI(C18)) in thin films is schematically shown in Fig. 1d. In these DCDA polymers, the CT state properties relevant for charge generation and recombination are determined by the intramolecular interfaces between the PBDB-T and NDI units. Therefore, the relevant DA spacing is related to the $d$-spacing, with the actual value of the DA spacing being considerably smaller than half of the $d$-spacing, due to the space taken by the side chains on the acceptor units (Fig. 1d). The $d$-spacing of the lamellae is expected to depend on the length of the alkyl linkers. Indeed,

from the GIWAXS patterns (Supplementary Note 3) of the thin films of the DCDA polymers with shorter linkers, i.e., PBDB-T-NDI(C6) and PBDB-T-NDI(C12), we find that the structural properties are similar to that with longer linkers, and the lamellar packing remains. Also, we derive that the $d$-spacings are 3.6 and 4.6 nm, for PBDB-T-NDI(C6) and PBDB-T-NDI(C12), respectively, considerably shorter than that for PBDB-T-NDI(C18) (5.2 nm). This suggests a reliable tuning of the DA spacing by a decrease of the linker length in the ordered regions of the thin DCDA polymer films.

Molecular dynamic simulations (MD) are also employed mainly to investigate the spatial arrangement of the donor and the acceptor units in the DCDA polymers and determine the DA spacing in the amorphous parts of the films. As shown in Supplementary Fig. 14, the MD simulations predict that the NDI acceptor units of PBDB-T-NDI(C6) are restricted to the close vicinity of the donor backbones by the short alkyl chain linkers, facilitating the formation of intramolecular parallel π-stacking between the donor backbones and the acceptor units. This is the closest packing structure that one can expect for the packed aromatic rings with small spacing (or DA spacing as is for the stacked donor and acceptor molecules)[65,66]. On the contrary, the NDI acceptor units of the DCDA polymer with longer alkyl chain linkers, e.g., PBDB-T-NDI(C18), are pushed away from the donor backbone by the linker groups (Supplementary Fig. 14), due to the large size of the linker group occupying considerable space around the donor backbones. This significantly reduces the probability for the parallel π stacking between the donor and the acceptor units. Using the centroids to represent the locations of the conjugated structures in the donor and the acceptor units, we derive the radial distribution function (RDF) for the distances between the donor and the acceptor units (Supplementary Fig. 14): For PBDB-T-NDI(C6), the peak value of the shortest

**Table 1 Voltage loss values from the subgap EQE measurements for the solar cells based on DCDA polymers.**

| | $V_{OC}$ (V) | $E_{CT}$ (eV) | $V_{loss}$[a] (V) | $V_{OC,rad}$[b] (V) | $\Delta V_{nr}$[c] (V) | $\Delta V_r$[d] (V) |
|---|---|---|---|---|---|---|
| PBDB-T-NDI(C6) | 0.66 | 1.48 | 0.82 | 1.22 | 0.56 | 0.26 |
| PBDB-T-NDI(C12) | 0.76 | 1.49 | 0.73 | 1.25 | 0.49 | 0.24 |
| PBDB-T-NDI(C18) | 0.81 | 1.51 | 0.70 | 1.25 | 0.44 | 0.26 |

JV curves and the photovoltaic performance parameters of the solar cells are provided in Supplementary Note 9.
[a]Calculated using eq. $V_{loss} = E_{CT}/q - V_{OC}$, the method used to determine $E_{CT}$ is provided in Supplementary Note 5.
[b]Calculated from the radiative limit for the dark saturation current, details in Supplementary Note 6.
[c]Calculated using eq. $\Delta V_{nr} = V_{OC,rad} - V_{OC}$; $\Delta V_{nr}$ values are also verified by measuring EQE$_{EL}$, details in Supplementary Note 6.
[d]Calculated using eq. $\Delta V_r = V_{loss} - \Delta V_{nr}$.

distances between the donor and the acceptor units, corresponding to the DA configuration with the closest spacing (most relevant for CT state formation), is 3.7 Å, and for PBDB-T-NDI(C18), the peak value is 4.2 Å. This shows that increasing the linker length indeed allows a tuning of the DA spacing also in the amorphous parts of the thin films of DCDA polymers.

**CT state decay rates and $V_{loss}$ in the DCDA solar cells**. With tunable DA spacing in the thin films established, we can now evaluate its influence on key photovoltaic and recombination parameters in organic solar cells. The devices are based on an inverted architecture of ITO/ZnO/active layer/MoO$_3$/Ag. We find that the $V_{OC}$ of the solar cells strongly depends on linker length, as listed in Table 1: The $V_{OC}$ of the solar cell based on PBDB-T-NDI(C6) with short linkers is 0.66 V, considerably lower than that based on PBDB-T-NDI(C18) with longer linkers (0.81 V). We also note that the $E_{CT}$ of the solar cells, determined using the method described in the literature[67–70], are very similar regardless of the length of the linkers (Fig. 2). Thus, $V_{loss}$, being the difference between $E_{CT}/q$ and the measured $V_{OC}$, is significantly higher for the solar cell based on PBDB-T-NDI(C6) (0.82 V), compared to that based on PBDB-T-NDI(C18) (0.70 V), as indicated in Fig. 2 and summarized in Table 1. More importantly, using the photovoltaic EQE, we determine the radiative limit for $V_{OC}$ ($V_{OC,rad}$). Then we calculate $\Delta V_{nr}$ ($\Delta V_{nr} = V_{OC,rad} - V_{OC}$)[31,71] and find that $\Delta V_{nr}$ is also higher in the solar cell based on PBDB-T-NDI(C6) (0.56 V), as compared to PBDB-T-NDI(C18) (0.44 V). On the other hand, the values of $\Delta V_r$ are similar in these devices, which suggests similar $k_r$[29,41,44]. Note that these results do not suggest that $k_r$ is completely independent of the DA spacing: There could be a small reduction in $k_r$ with the increased DA spacing, but this reduction could not be detected by the measurement of $\Delta V_r$, due to the logarithmic dependence of $\Delta V_r$ on $k_r$. Furthermore, the values for $V_{OC,rad}$ are verified by measuring the EQE$_{EL}$ of the devices, details regarding the determination of the $V_{loss}$ terms are provided in Supplementary Note 6.

From the above results, the increase in EQE$_{EL}$ and decrease in $\Delta V_{nr}$ with increasing chain length must therefore be mainly due to a decreased $k_{nr}$ (EQE$_{EL} \approx k_r/k_{nr}$). This is confirmed by the transient photovoltage decay (TPV) measurements: The voltage decay time, i.e., the lifetime of the photo-generated charge carriers, closely related to the inverse of CT state decay rate ($k_r + k_{nr} \approx k_{nr}$), is approximately an order of magnitude higher for the solar cell based on PBDB-T-NDI(C18), compared to that based on PBDB-T-NDI(C6) (Details regarding the TPV measurements are given in Supplementary Note 7). These results provide strong evidence that an increased DA spacing, giving rise to a reduced vibrational coupling between the donor and the acceptor material, leads to reduced $k_{nr}$ and suppressed $V_{loss}$.

Importantly, we expect that a larger DA spacing than in the PDBD-T-NDI(C18) case would be desirable. Because despite having the largest DA distance and the lowest $V_{loss}$, the photoluminescence (PL) of S$_1$ states of PDBD-T-NDI(C18) is

still completely quenched (Supplementary Fig. 22), indicating that electron transfer is highly efficient[72–75]. Besides, we find that the peak values of the photovoltaic EQE of the solar cells based on the longer linkers are not lower, as compared to that based on the shorter linkers (Supplementary Fig. 22). Therefore, further increasing of the linker length in the DCDA polymer is expected to increase the DA spacing, and thus, to reduce $k_{nr}$ and improve the performance of the solar cell, although the synthesis of such a material is highly challenging, due to the poor solubility of the raw material needed.

**Impact of increased DA spacing on $V_{loss}$ in BHJ solar cells**. We now evaluate whether typical state-of-the-art BHJ solar cells based on non-chemically linked donor and acceptor materials would also benefit from a larger DA spacing. We vary the size of the donor material's side chains and focus on their role in determining the DA spacing in the BHJ systems and the performance of solar cells. The reference BHJ systems are based on the classic donor polymer PBDB-T with 16 hydrocarbyl groups attached to the BDT (alkylthienyl-substituted benzo[1,2-b:4,5-b′] dithiophene) moiety[3] mixed with a series of different high-performance non-fullerene acceptors, including IT4F[7], ITIC[5], and Y6[8]. The reference systems are compared with the systems based on the PBDB-T(OD) donor having 40 hydrocarbyl groups on the BDT moiety. The chemical structures of the donor materials and the non-fullerene acceptors are shown in Fig. 3a.

The impact of the change of the side chain size on $V_{loss}$ in the BHJ solar cells is very similar to the results observed from the solar cells based on DCDA polymers with varying size of linker groups, as listed in Table 2: Regardless of the acceptor used, the solar cells based on PBDB-T(OD) have significantly lower $V_{loss}$, as compared to those based on PBDB-T, due to lower $\Delta V_{nr}$, associated with lower $k_{nr}$ (confirmed by TPV, shown in Fig. 4c and Supplementary Fig. 21). Thus, the lower $V_{loss}$ in the solar cells based on PBDB-T(OD) is also associated with larger DA spacing.

Indeed, from MD simulations, we find that the DA spacing is considerably larger in the blends based on PBDB-T(OD). More specifically, for the blend based on PBDB-T, for instance, PBDB-T:IT4F (Fig. 3b), the side chains of PBDB-T stretch out, leaving the conjugated backbone fully exposed, and highly accessible to the acceptor molecules. This promotes an efficient formation of the parallel π-stacking between PBDB-T and the acceptors. On the contrary, in the blend based on PBDB-T(OD) (Fig. 3c), the large side chains of PBDB-T(OD) fold back to the conjugated backbone, taking up space around the backbone. This makes it highly difficult for the formation of close packing between PBDB-T(OD) and the acceptors. Thus, the distance between the conjugated donor backbones and the acceptor molecules is considerably larger in the blend based on PBDB-T(OD) than that based on PBDB-T. The RDF (Fig. 3d) shows a peak value for the DA configuration with the closest spacing at 3.9 Å for PBDB-T:IT4F, which increases to 4.8 Å for PBDB-T(OD):IT4F. Similar

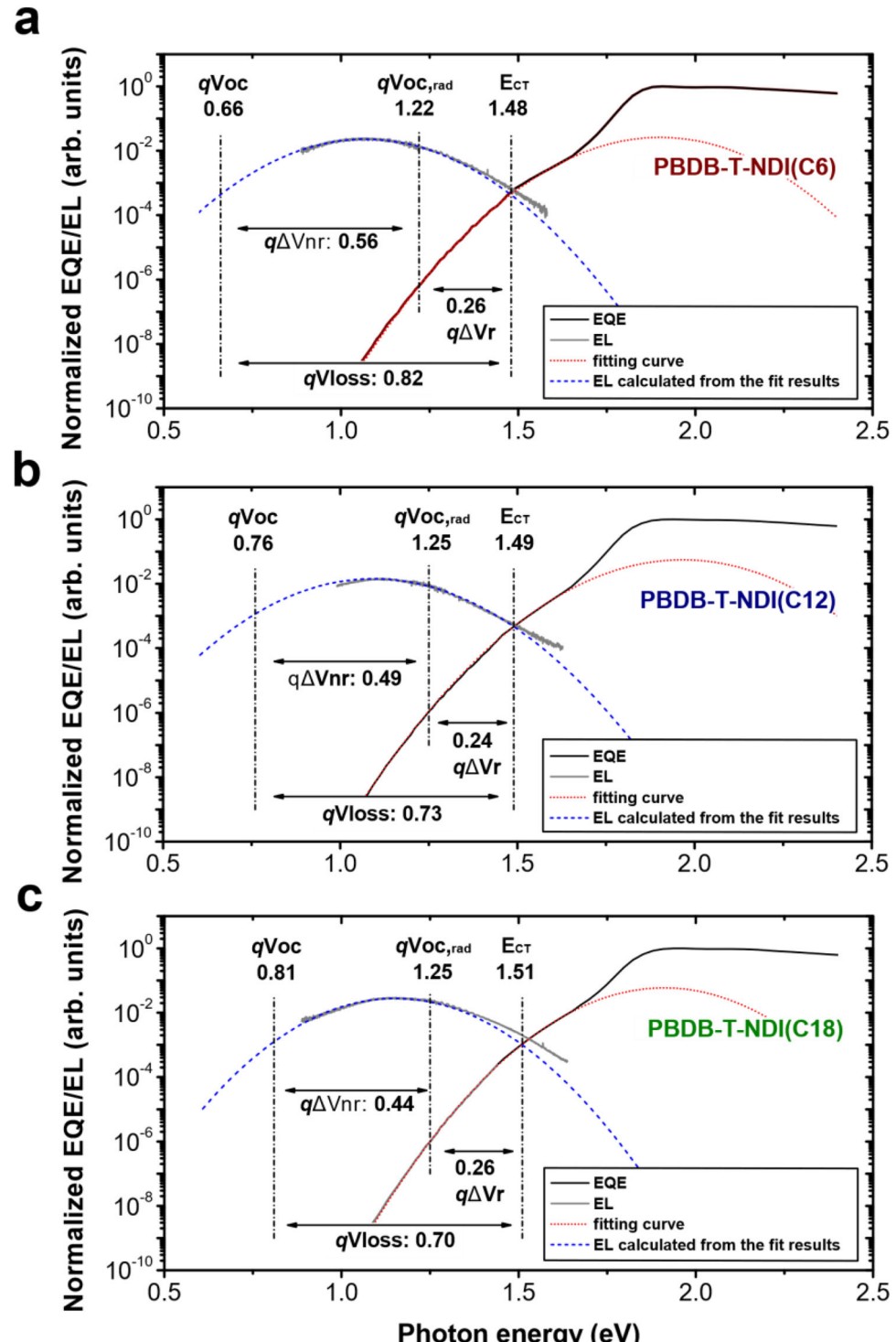

**Fig. 2 Voltage losses and CT state properties in the solar cells based on DCDA polymers.** EQE and electroluminescence (EL) spectra of the solar cells based on **a** PBDB-T-NDI(C6), **b** PBDB-T-NDI(C12), and **c** PBDB-T-NDI(C18). The lower energy parts of the EQE spectra are fitted by using the method described in the literature[67–70] to determine the values for $E_{CT}$, and the fit results are used to calculate the emission spectra, which agree well with the measured EL spectra of the solar cells. Details regarding the determination of $E_{CT}$ are provided in Supplementary Note 5.

simulation results are found using ITIC or Y6 as the acceptor material (Supplementary Note 4), suggesting that the DA spacing is always larger in the blends based on PBDB-T(OD) with larger side chains.

The above results suggest that the DA spacing can also be increased in BHJ organic solar cells comprising state-of-the-art materials by using long side chains. Also in these devices, the

increased DA distance leads to lower $k_{nr}$ and higher $V_{OC}$, due to the reduced degree of vibrational coupling between CT and ground states. Since the PL quenching of both the donor and the acceptor $S_1$ states is highly efficient in the blends based on PBDB-T(OD), suggesting a highly efficient electron transfer process in these blends, despite having the DA spacing up to 5 Å (Supplementary Fig. 23). Accordingly, a further increased DA

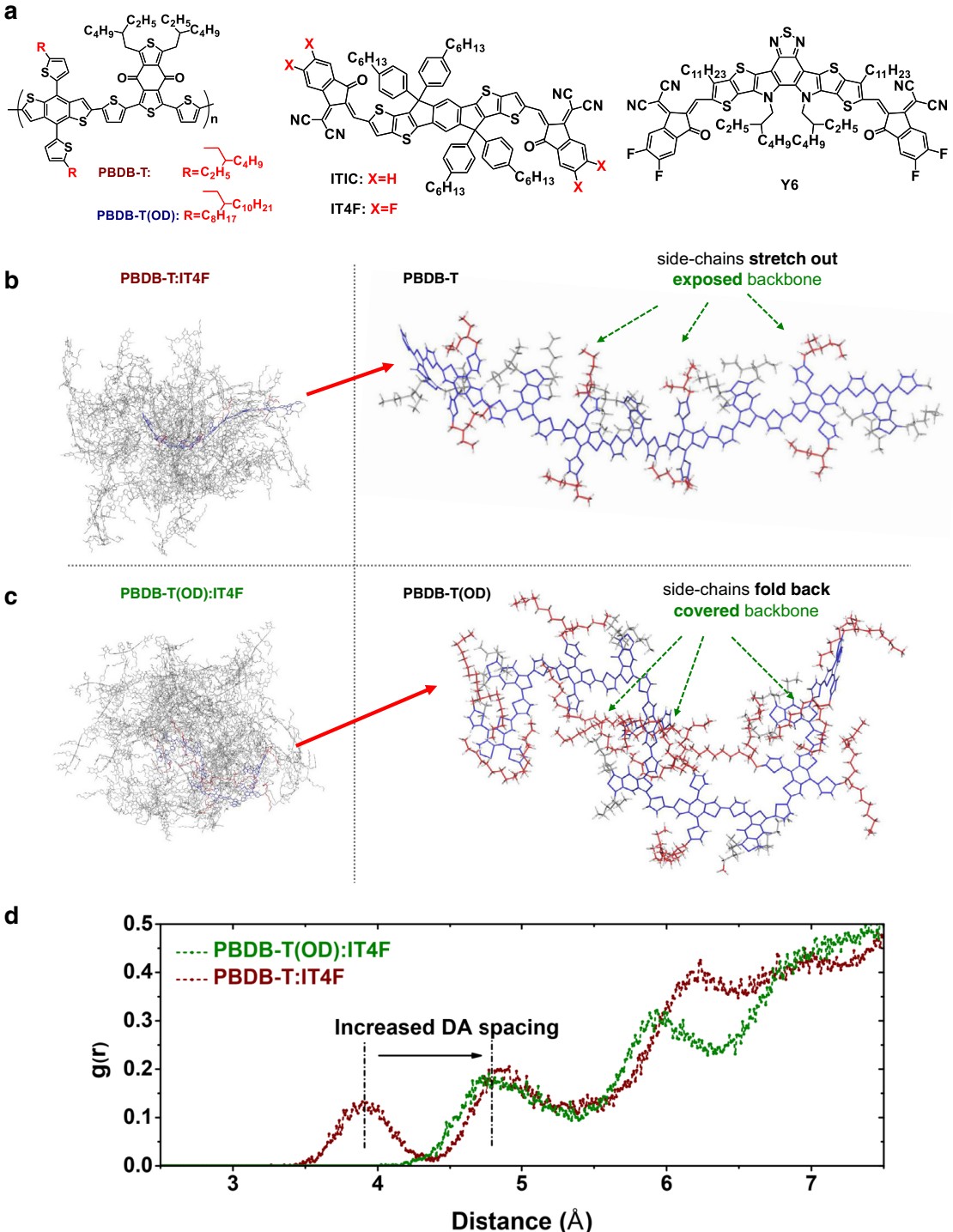

**Fig. 3 Molecular dynamic simulations for the BHJ systems. a** Chemical structures of PBDB-T, PBDB-T(OD), and the acceptor materials used. MD simulation results for **b** PBDB-T and **c** PBDB-T(OD) blended with IT4F. The side chains of PBDB-T stretch out, leaving the conjugated backbones highly accessible to IT4F, allowing for close parallel π stacking between the backbones and IT4F, while the side chains of PBDB-T(OD) fold back, covering the backbones, reducing the probability for IT4F to stack on the donor backbones. **d** RDF of the distances between the donor backbones and IT4F for the blends based on PBDB-T and PBDB-T(OD).

spacing is expected to lead to a higher degree of reduction in $k_{nr}$ and a lower $V_{loss}$.

Inspired by the significant reduction in $V_{loss}$, achieved by modifying the side chains of the donor material, we now employ a new acceptor, Y6(OD) (Fig. 4a), with side chains being significantly larger than that of Y6, to construct solar cells.

Remarkably, the $\Delta V_{nr}$ of the solar cell based on PBDB-T(OD):Y6(OD) is lower by as much as 0.08 V, compared to that based on PBDB-T(OD):Y6 (Fig. 4b), and the lower $\Delta V_{nr}$ is ascribed to a lower $k_{nr}$, confirmed by TPV measurements (Fig. 4c): The voltage decay lifetime in the solar cell based on PBDB-T(OD):Y6(OD) is almost two orders of magnitudes longer

**Table 2 Voltage loss values from the subgap EQE measurements for the BHJ solar cells based on PBDB-T and PBDB-T(OD) mixed with different acceptors.**

|  | $V_{OC}$ (V) | $E_{CT}/E_g{}^a$ (eV) | $V_{loss}{}^b$ (V) | $V_{OC,rad}{}^c$ (V) | $\Delta V_{nr}{}^d$ (V) | $\Delta V_r{}^e$ (V) |
|---|---|---|---|---|---|---|
| PBDB-T:IT4F | 0.71 | 1.40 | 0.69 | 1.15 | 0.44 | 0.25 |
| PBDB-T(OD):IT4F | 0.83 | 1.45 | 0.62 | 1.20 | 0.37 | 0.25 |
| PBDB-T:ITIC | 0.90 | 1.52 | 0.62 | 1.24 | 0.34 | 0.28 |
| PBDB-T(OD):ITIC | 0.99 | 1.56 | 0.57 | 1.27 | 0.28 | 0.29 |
| PBDB-T:Y6 | 0.67 | 1.42 | 0.75 | 1.04 | 0.37 | 0.38 |
| PBDB-T(OD):Y6 | 0.77 | 1.42 | 0.65 | 1.06 | 0.29 | 0.36 |
| PBDB-T(OD):Y6(OD) | 0.86 | 1.41 | 0.55 | 1.07 | 0.21 | 0.34 |

JV curves and the photovoltaic performance parameters of the solar cells are provided in Supplementary Note 9.
a$E_g$ instead of $E_{CT}$ are listed for the solar cells based on Y6 and Y6(OD), details in Supplementary Note 5.
bCalculated using eq. $V_{loss} = E_{CT}/q - V_{OC}$, the method used to determine $E_{CT}$ is provided in Supplementary Note 5.
cCalculated from the radiative limit for the dark saturation current, details in Supplementary Note 6.
dCalculated using eq. $\Delta V_{nr} = V_{OC,rad} - V_{OC}$; $\Delta V_{nr}$ values are also verified by measuring $EQE_{EL}$, details in Supplementary Note 6.
eCalculated using eq. $\Delta V_r = V_{loss} - \Delta V_{nr}$.

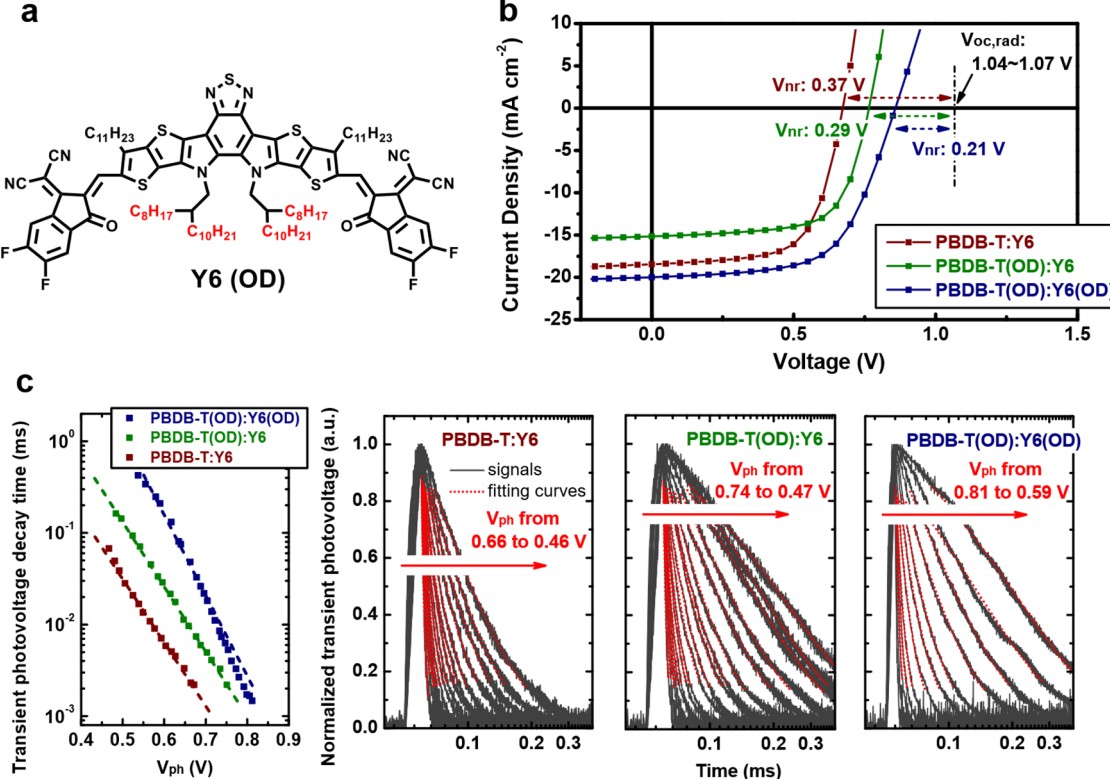

**Fig. 4 Impact of side-chain modification on the performance of BHJ organic solar cells. a** Chemical structure of Y6(OD) with side chains considerably large than Y6. **b** JV curves of the solar cells based on PBDB-T, PBDB-T(OD), mixed with Y6 and Y6(OD). $\Delta V_{nr}$ values, calculated from $V_{OC,rad}$, are indicated. **c** TPV measurements performed on the solar cells based on the active materials with and without the side chain modification, the voltage decay lifetime values (left) are derived from the transient signals (right) of the solar cells measured under different steady-state illumination intensities (to generate different steady-state photovoltage, ($V_{ph}$)). The dash lines in the left plot are a guide to the eyes, and details regarding the measurements are provided in Supplementary Note 7.

than that based on PBDB-T:Y6. This leads to a further increase of $V_{OC}$ to up to 0.86 V, close to 0.2 V higher than that of the solar cell based on unmodified PBDB-T:Y6.

## Discussion

The currently highest PCEs in organic solar cells over 18%[9,10,16,17,65,76] have been achieved with solution cast mixtures, without thorough consideration of the intermolecular DA spacing. Very few methods to quantify and control this spacing are available, and it has therefore been unclear if higher efficiencies can be reached by careful manipulation of the DA spacing, even

though there are indications that an optimum distance, allowing for efficient photo-induced electron transfer but suppressed recombination of the CT state, exists: Indeed, in the analogous and highly efficient primary photo-induced long-range (tens of Å) electron transfer processes in photosynthesis, recombination of the charge-separated state is severely suppressed, the molecular orientation and distance of the special donor-acceptor pair in the reaction center is well defined and controlled by proteins holding the molecules in place[47,66,77–79].

In this work, we achieved tuning of the DA spacing in both DCDA and BHJ organic solar cells, and observed a decreased

non-radiative recombination and increased $V_{OC}$ with increased DA spacing, while the increased spacing had little impact on the dissociation efficiency of the excitons at the donor-acceptor interface. Therefore, we concluded that all of the active materials systems studied in this work, and by extension, the current systems studied for organic solar cells in the literature could benefit from a larger DA spacing. We hope that the conclusion, i.e., the DA spacing in organic solar cells is generally too small, being the main reason for the too high $k_{nr}$ in organic solar cells, will inspire researchers to focus on innovative methods to manipulate the DA spacing, and we envision that these methods, such as engineering the non-conjugated part of the active materials demonstrated here, will be of great importance for the future development of organic solar cells.

## Methods

**Statistics and reproducibility**. All the characterizations and measurements performed in this work were repeated by different persons, and similar results were obtained.

**Experimental**. Details regarding the methods used in this work are provided in supporting information.

**Reporting Summary**. Further information on research design is available in the Nature Research Reporting Summary linked to this article.

## Data availability

The data that support the findings of this study are available from the corresponding authors.

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

## Acknowledgements

This work is financially supported by the National Natural Science Foundation of China (Grant No. 51973031, 51933001, 52073056, and 52073016), Shanghai Pujiang Program (Grant No. 19PJ1400500), the Natural Science Foundation of Shanghai (Grant No. 19ZR1401400), the Fundamental Research Funds for the Central Universities (Grant No. 2232021A09 and XK1802-2), the MOST (2017YFA0204702 and 2018YFA0208504), the Jiangxi Provincial Department of Science and Technology (No. 20192ACB20009), and the European Research Council (ERC, grant No. 864625). The authors thank Prof. Olle Inganäs for the discussion and his comments on the manuscript.

## Author contributions

J.W. and X.J. contributed equally to this work; This project was designed by Z.T. and supervised by Z.T., W.L., and K.V.; J.W., X.J., and G.F. fabricated and optimized the devices; X.J. synthesized the DCDA polymers; J.W. performed JV, EQE, EQE$_{EL}$, EL, and TPV measurements, supervised by Z.M.; HY.W. and X.F. measured TEM; HB.W. and Y.Y. did MD simulations; J.L. measured GIWAXS; Z.T. wrote the manuscript with K.V., and all authors contributed to the discussion and the finalizing of the manuscript.

## Competing interests

The authors declare no competing interests.
