## [Peer Review File · Nature Communications]

Increasing donor-acceptor spacing for reduced voltage loss in organic solar cellsREVIEWER COMMENTS

Reviewer #1 (Remarks to the Author):

In this work, the authors carried out a careful study on the effect of DA spacing on the non-radiative recombination of organic solar cells. Results show that in typical organic solar cells, the DA spacing is generally too small, which induces the too-fast non-radiative decay (k_{nr}) of charge carriers and large ΔV_{nr} . Therefore, V_{loss} can in fact be suppressed by controlling the spacing between the D and A materials, e.g. alkyl chain spacers in single component DA systems and side chains in high-efficiency bulk-heterojunction systems. This work points out a new viewpoint for breaking the performance bottleneck of organic solar cells. I recommend the publication of this work after the following questions are addressed.

1. The author claimed that the peak values of the photovoltaic EQE of the solar cell based on PBDB-T-NDI(C18) is similar to that of the solar cells based on the DCDA polymers with shorter linkers (Figure S8.1b). But the peak EQE values in Figure S8.1b vary from 40% to 60%. Should these values be viewed as similar?
2. The morphology characterization of BHJ devices should be given.
3. In figure S1.3.b, PBDB-T-NDI(12) shows the lowest absorption coefficient among these three polymers. However, it delivers the highest PCE and J_{sc} . Can the author make a discussion on this phenomenon as the charge transfer of these polymers are all efficient and the driving force are comparable (estimated from $(E_{gCV-ECT})/q$).
4. The written English should be polished. Besides, in figure 4, the n_r and p_h in ΔV_{nr} , V_{ph} should be subscript.

Reviewer #2 (Remarks to the Author):

This manuscript deals with organic solar cells. The main finding discussed in this work is that the voltage loss can be decreased by increasing the separation between donor (D) and acceptor (A) units.

In order to demonstrate their point the authors investigated solar cells based on DCDA polymers in which the PBDB-T donor units and NDI acceptor units are linked using alkyl chain with different numbers (N) of methylene groups.

The results of TEM and GIWAXS measurements as well as the results of MD simulations show indeed that with increase in N the average D-A distance increases. The EL measurements show that the non-radiative voltage loss decreases from 0.56 eV to 0.44 eV when N is increased from 6 to 18. The authors concluded that this trend is due to the decrease in the non-radiative decay rate.

The problem addressed in this work is of great interest. However, before the paper could be recommended for publication, the authors should address the comments given below:

(1) The conclusion that the decrease in the non-radiative voltage loss is due to non-radiative recombination is not directly supported by the data. The measurement of both radiative and non-radiative recombination rates is needed.

(2) My main concern, however, is related to the interpretation of the results. First, the authors make use of the Eq.1 to suggest that the decrease in the non-radiative rate is because of the decrease in the electronic coupling when D-A separation increases. However, previous work by Koen Vandewal (Nat. Energy 2,17053 (2017)) has demonstrated that the change in the electronic coupling has no effect on the ratio of the non-radiative and radiative rates and, consequently, on the non-radiative voltage loss. Then on page 7 and in the abstract is mentioned (without any evidence presented) that the reduction of the non-radiative rate is due to the vibrational coupling between the donor and the acceptor material. So, is the electronic coupling or vibrational coupling responsible for the observed effect? How exactly?

(3) The effect of D-A distance on device performance has been previously investigated in the context of exciplex-TADF systems, see for instance Mater. Horiz., 2021, 8, 401 (DOI: 10.1039/D0MH01245A). The authors might consider including these earlier findings in their discussion.

Reviewer #3 (Remarks to the Author):

The authors propose a very relevant concept that is of high interest to the OPV community. While Voc losses are often related to the energetics of the donor:acceptor (D:A) system, the electronic coupling to the ground state (which impacts the rate of non-radiative recombination) is often overlooked. Here, it is shown that non-radiative recombination and Voc losses can be reduced by increasing the D-A distance (i.e. lowering the coupling). This is shown both for a covalently linked double cable D-A polymer and for typical bulk heterojunctions (BHJs). While the message is very impactful, it is not yet sufficiently substantiated to be convincing. Therefore, I suggest publication in Nature Communications after the following revisions:

1) Apart from modulating the D-A distance, the changes in packing due to the linkers or side chains could also affect the aggregation/delocalization in the donor and acceptor domains (e.g. delocalization of charges can occur in NDI stacks). How does this affect the charge generation and recombination dynamics (and voltage losses)?

2) Can the authors explain why the radiative recombination rate does not depend on the electronic coupling/D-A distance?

3) The transient voltage decay measurements should be explained in more detail (information in the SI is not enough). Why does the voltage decay time represent the lifetime of the photogenerated charges (i.e. the CT decay rate)? Does it not rather depend on extraction and free charge lifetime?

4) There is not enough structural characterization (e.g. GIWAXS) for the BHJ blends, only the MD simulations are not convincing.

5) Apart from the D-A distance, the coupling also depends on the orientation of the molecules. How is this controlled in the studied systems?

Thank you for considering our manuscript for potential publication in *Nat. Commun.* We would also like to thank the reviewers for their overall positive assessment and constructive comments on our manuscript. Below is our response to the reviewers' comments, and the changes made in the revised manuscript are marked in red.

Yours sincerely,
On behalf of all authors,
Weiwei Li, Koen Vandewal, and Zheng Tang

REVIEWER COMMENTS

Reviewer #1 (Remarks to the Author):

In this work, the authors carried out a careful study on the effect of DA spacing on the non-radiative recombination of organic solar cells. Results show that in typical organic solar cells, the DA spacing is generally too small, which induces the too-fast non-radiative decay (k_{nr}) of charge carriers and large ΔV_{nr} . Therefore, V_{loss} can in fact be suppressed by controlling the spacing between the D and A materials, e.g. alkyl chain spacers in single component DA systems and side chains in high-efficiency bulk-heterojunction systems. This work points out a new viewpoint for breaking the performance bottleneck of organic solar cells. I recommend the publication of this work after the following questions are addressed.

Our response: We appreciate the comments from the reviewer.

1. The author claimed that the peak values of the photovoltaic EQE of the solar cell based on PBDB-T-NDI(C18) is similar to that of the solar cells based on the DCDA polymers with shorter linkers (Figure S8.1b). But the peak EQE values in Figure S8.1b vary from 40% to 60%. Should these values be viewed as similar?

Our response: This statement (in **Page 8**) is now revised as “we find that the peak values of the photovoltaic EQE of the solar cells based on the longer linkers are not lower as compared to those based on the shorter linkers”.

2. The morphology characterization of BHJ devices should be given.

Our response: We now have included TEM images and GIWAXS results for the BHJ systems studied in this work. Please see **SI-2** and **SI-3** of the revised manuscript.

3. In figure S1.3.b, PBDB-T-NDI(12) shows the lowest absorption coefficient among these three polymers. However, it delivers the highest PCE and J_{sc} . Can the author make a discussion on this phenomenon as the charge transfer of these polymers are all efficient and the driving force are comparable (estimated from $(E_g - E_{CT})/q$).

Our response: To understand the reason for the different J_{sc} of the solar cells based on the different DCDA polymers, transfer matrix model (TMM) simulations, done using the measured dielectric functions of the materials used in the solar cells (*J. Appl. Phys.* 1999, 86, 487), are now included in the revised manuscript. The results from the TMM simulations

are shown **Figure R1**.

Figure R1. J_{sc} simulated by transfer matrix simulations for the DCDA solar cells with different active layer thicknesses. The optical constants of the materials used in the solar cells are measured by spectroscopic ellipsometry.

We find that the J_{sc} predicted by the TMM are similar for all of the DCDA solar cells. This suggests that the difference in the measured J_{sc} of the solar cells is a result of different device internal quantum efficiencies (IQEs). For organic solar cells, IQE is primarily determined by the dissociation efficiency of the S_1 state into free charge carriers and their extraction efficiency. Since we found from the PL measurements (**Figure S8.1**) that the S_1 dissociation efficiency is high (over 90%) in all of the DCDA polymer based films, the lower IQE of the solar cell based on PBDB-T-NDI(C6) with the shortest linker, as compared to that based on PBDB-T-NDI(C12) and PBDB-T-NDI(C18), is most likely due to less efficient charge carrier extraction. This could also explain the strongly field dependent photocurrent extraction in the solar cell based on PBDB-T-NDI(C6) (**Figure S9.1**). In this work, we demonstrate that the decay rate of CT state is very high for the solar cell based on PBDB-T-NDI(C6), it is thus reasonable to attribute the rather inefficient photo-conversion to the high decay rate of the CT state, resulting in a comparably low IQE and low J_{sc} in the solar cell based on PBDB-T-NDI(C6).

For the solar cells based on PBDB-T-NDI(C12) and PBDB-T-NDI(C18), the extraction of photocurrent is independent of electric field (**Figure S9.1**). This suggests that in PBDB-T-NDI(C12) and PBDB-T-NDI(C18), the CT state decay rate, which is reduced as compared to that of PBDB-T-NDI(C6), does not limit the J_{sc} . Also, it suggests that the slightly lower J_{sc} of the solar cell based on PBDB-T-NDI(C18), as compared to that based on PBDB-T-NDI(C12), is due to a less efficient dissociation of excitons. PL measurements (**Figure S8.1**) reveals that quenching of the acceptor emission is indeed less efficient in the active layer based on PBDB-T-NDI(C18), as compared to that based on PBDB-T-NDI(C12). This could be a result of a relatively higher degree of aggregation of the acceptor units in the film of PBDB-T-NDI(C18), due to the higher degree of spatial freedom of the acceptor units in the DCDA polymer with longer linkers.

In **SI-9** of the revised manuscript, a brief discussion regarding the origin of the different

J_{sc} in the solar cells based on the different DCDA polymers is added.

4. The written English should be polished. Besides, in figure 4, the nr and ph in ΔV_{nr} , V_{ph} should be subscript.

Our response: We have carefully corrected the linguistic problems for the revised manuscript.

Reviewer #2 (Remarks to the Author):

This manuscript deals with organic solar cells. The main finding discussed in this work is that the voltage loss can be decreased by increasing the separation between donor (D) and acceptor (A) units. In order to demonstrate their point the authors investigated solar cells based on DCDA polymers in which the PBDB-T donor units and NDI acceptor units are linked using alkyl chain with different numbers (N) of methylene groups. The results of TEM and GIWAXS measurements as well as the results of MD simulations show indeed that with increase in N the average D-A distance increases. The EL measurements show that the non-radiative voltage loss decreases from 0.56 eV to 0.44 eV when N is increased from 6 to 18. The authors concluded that this trend is due to the decrease in the non-radiative decay rate.

The problem addressed in this work is of great interest. However, before the paper could be recommended for publication, the authors should address the comments given below:

Our response: We appreciate the comments from the reviewer.

(1) The conclusion that the decrease in the non-radiative voltage loss is due to non-radiative recombination is not directly supported by the data. The measurement of both radiative and non-radiative recombination rates is needed.

Our response: In this manuscript, the reduced non-radiative voltage loss (ΔV_{nr}) in the solar cell with larger DA distances, compared to that with smaller DA distances, is associated the reduced non-radiative decay rate of CT state (k_{nr}). This conclusion is drawn from the results of two different sets of experiments.

1) Non-radiative voltage loss (ΔV_{nr}) is inversely proportional to the EQE_{EL} of the solar cell,

$$\Delta V_{nr} = \frac{kT}{q} \ln\left(\frac{1}{EQE_{EL}}\right) \quad (\text{eq. R1})$$

Thus, the reduced ΔV_{nr} in the solar cells with larger DA distances must be associated with increased EQE_{EL} . This is confirmed by the EQE_{EL} measurements (**Table S6.1**). Furthermore,

$$EQE_{EL} = \frac{k_r}{k_r + k_{nr}} \approx \frac{k_r}{k_{nr}} \quad \text{for } EQE_{EL} \ll 1 \quad (\text{eq. R2})$$

where k_r is the radiative decay rate of CT state. Thus, the increased EQE_{EL} could be either due to increased k_r or reduced k_{nr} . However, the increased k_r should lead to increased radiative voltage loss (ΔV_r) of the solar cell, since (*Annu. Rev. Phys. Chem.*

$$\Delta V_r \propto \frac{kT}{q} \ln(k_r) \quad (\text{eq. R3})$$

But this is not the case for the solar cells with different DA distances studied in this work (**Table 1**). Therefore, k_{nr} must be reduced in the solar cells with larger DA distances.

- 2) We also perform transient photovoltage decay (TPV) measurements, and compare the voltage decay lifetime of the solar cells with different DA distances: We observe that the voltage decay lifetime is longer in the solar cells with larger DA distances, implying that k_{nr} is lower. Thus, we are convinced that the increased EQE_{EL} , and thus the reduced ΔV_{nr} , in the solar cells with increased DA distances is due to the reduced k_{nr} . In **SI-7** of the revised manuscript, a more detailed discussion regarding the TPV measurements, and the dependence of voltage decay lifetime on k_{nr} in organic solar cells is provided.

We agree that a direct measurement of k_r and k_{nr} of the CT states would help us further strengthening the conclusion of the manuscript. However, such a kinetic measurement, able to distinguish k_r and k_{nr} , is not possible to the best of our knowledge. Kinetic measurements, including transient absorption spectroscopy, will also measure the lifetime of the charge carriers, which depends on several rates, including k_r and k_{nr} , but also the CT state dissociation rate and the electron-hole encounter rate.

(2) My main concern, however, is related to the interpretation of the results. First, the authors make use of the Eq.1 to suggest that the decrease in the non-radiative rate is because of the decrease in the electronic coupling when D-A separation increases. However, previous work by Koen Vandewal (*Nat. Energy* 2,17053 (2017)) has demonstrated that the change in the electronic coupling has no effect on the ratio of the non-radiative and radiative rates and, consequently, on the non-radiative voltage loss. Then on page 7 and in the abstract is mentioned (without any evidence presented) that the reduction of the non-radiative rate is due to the vibrational coupling between the donor and the acceptor material. So, is the electronic coupling or vibrational coupling responsible for the observed effect? How exactly?

Our response: In the paper published in 2017 (*Nat. Energy*, 2,17053, 2017), the expressions used to describe k_r and k_{nr} of the CT states are

$$k_{nr} \propto V^2 FC(g = E_{CT}) \quad (\text{eq. R4})$$

$$\frac{k_r(\nu)}{\nu} \propto V^2 \Delta\mu^2 FC(g = E_{CT} - h\nu) \quad (\text{eq. R5})$$

These expressions suggest that k_{nr} does depend on the electronic coupling, V , but the ratio of $\frac{k_r}{k_{nr}}$ seems to be independent of V . However, it should be noted that eq. R5 is highly simplified.

In the paper published by Azzouzi et al. in 2018 (*Phys. Rev. X*, 2018, 8, 031055), it

has been derived that k_r is related to the transition dipole moment (M),

$$k_r(\hbar\omega) = \frac{1}{3\pi\epsilon_0\hbar^2} \left(\frac{\hbar\omega}{c}\right)^3 M^2 FCWD(\hbar\omega) \quad (\text{eq. R6})$$

and

$$M^2 = \frac{3}{2} \frac{\hbar^2}{\hbar\omega_{avg} m_e} f_{osc} \quad (\text{eq. R7})$$

Thus,

$$EQE_{EL} = \frac{k_r}{k_{nr}} \propto \frac{f_{osc}}{V^2} \quad (\text{for } k_{nr} \gg k_r) \quad (\text{eq. R8})$$

Therefore, EQE_{EL} , and thus the non-radiative voltage loss ΔV_{nr} , of the DA solar cells must be dependent on the electronic coupling parameter V : EQE_{EL} increases, and ΔV_{nr} reduces with reducing V .

Furthermore, using the generalized Mulliken-Hush method, V can be expressed as a function of M and $\Delta\mu^2$

$$V = \frac{ME_{CT}}{\sqrt{\Delta\mu^2 + 4M^2}} \quad (\text{eq. R9})$$

M reduces with increasing DA spacing, and $\Delta\mu^2$ increases with DA spacing. Thus, it can be concluded that increasing DA spacing leads to reduced V , and thus, increased EQE_{EL} and reduced ΔV_{nr} .

We now realize that the lack of the explanation of the different forms of the rate equations used for organic solar cells could lead to potential confusions. Thus, in the introduction of the revised manuscript (**Page 2 and 3**), the text regarding the important role the electronic coupling plays in determining ΔV_{nr} is modified for a better clarification.

(3) The effect of D-A distance on device performance has been previously investigated in the context of exciplex-TADF systems, see for instance Mater. Horiz., 2021, 8, 401(DOI: 10.1039/D0MH01245A). The authors might consider including these earlier findings in their discussion.

Our response: This reference about the impact of DA spacing on LEDs based on exciplex-TADF is now cited and discussed in the introduction of the revised manuscript (**Page 3**).

Reviewer #3 (Remarks to the Author):

The authors propose a very relevant concept that is of high interest to the OPV community. While Voc losses are often related to the energetics of the donor:acceptor (D:A) system, the electronic coupling to the ground state (which impacts the rate of non-radiative recombination) is often overlooked. Here, it is shown that non-radiative recombination and Voc losses can be reduced by increasing the D-A distance (i.e. lowering the coupling). This is shown both for a covalently linked double cable D-A polymer and for typical bulk heterojunctions (BHJs). While the message is very impactful, it is not yet sufficiently substantiated to be convincing. Therefore, I suggest publication in Nature Communications after the following revisions:

Our response: We thank the reviewer for the comments.

1) Apart from modulating the D-A distance, the changes in packing due to the linkers or side chains could also affect the aggregation/delocalization in the donor and acceptor domains (e.g. delocalization of charges can occur in NDI stacks). How does this affect the charge generation and recombination dynamics (and voltage losses)?

Our response: For the solar cells studied in this work, we note that packing properties are not much affected by the change of the size of the linkers or the side chains. As can be seen from the TEM images (SI-2) and the GIWAXS results (SI-3). Thus, the reason for the reduced voltage losses is primarily assigned to the increased DA spacing.

However, we do agree with the reviewer that the property of molecular packing has significant impact on the recombination dynamics of charge carriers in the DA systems. For organic solar cells, both the radiative and the non-radiative decay rate of CT states, and thus the non-radiative voltage loss (ΔV_{nr}), are determined by the CT state related parameters. Azzouzi et al. derived that (*Phys. Rev. X*, 2018, 8, 031055)

$$\Delta V_{nr} = -kT \ln \left(p_e \frac{1}{3\pi\epsilon_0\hbar^4} \frac{(|\Delta\mu|^2 + 6\frac{\hbar^2}{\hbar\omega_{avg}m_e}f_{osc}) \int FCWD(\hbar\omega)(\hbar\omega)^3 d\hbar\omega}{\frac{2\pi}{\hbar}FCWD(0)E_{CT}^2} \right) \quad (\text{eq. R10})$$

where p_e is the emission probability of CT state, $\Delta\mu$ is the difference between the dipole moment of the CT and the ground state, f_{osc} is the oscillator strength of CT state, and $FCWD$ is the Franck-Condon weighted density of states, related to the reorganization energy (λ) and the vibrational modes of the CT and ground states.

These CT state related parameters, such as E_{CT} and λ , etc., are strongly dependent on the packing of the molecules. For instance, compared to a fully disordered system, a more ordered structure of the donor and the acceptor molecules is expected to result in lower reorganization energy (*J. Phys. Chem. C*, 118, 14848 (2014)), giving rise to reduced ΔV_{nr} .

Also, the molecular packing is one of the most important parameters determining the generation rate of CT states, and thus the efficiency of charge carrier generation in the DA systems: The probability of dissociation of the singlet excitons depends on the density of the charge transfer complex, which is dependent on the structural properties of the donor and the acceptor molecules in the DA system.

In the revised manuscript, it is further stressed that the DCDA polymers are used as the model system in this work, because they have rather stable and predictable packing properties (Page 5). This allows us to focus on the role the DA spacing plays in determining the decay dynamics of CT state.

2) Can the authors explain why the radiative recombination rate does not depend on the electronic coupling/D-A distance?

Our response: In the paper by Azzouzi et al. (*Phys. Rev. X*, 2018, 8, 031055), it has been

derived that

$$k_{nr} \propto V^2 \quad (\text{eq. R11})$$

$$k_r \propto M^2 \propto \frac{|\Delta\mu|^2}{\left(\frac{E_{CT}}{V}\right)^{-4}} \quad (\text{eq. R12})$$

Thus, k_r does depend on the electronic coupling, and thus the DA distance.

Because V^2 decreases with increasing DA spacing, k_{nr} should also decrease with increasing DA spacing (eq. R11). However, $|\Delta\mu|^2$ increases with DA spacing. Thus, k_r , depending on both $|\Delta\mu|^2$ and V^2 (eq. R12), does not necessarily decrease, or it does not decrease as rapidly as k_{nr} , with the increasing DA spacing.

In this manuscript, we calculate ΔV_r of the solar cells with different DA spacings, and observe that ΔV_r s are similar. This indicates that k_r s of the solar cells are not very different, since ΔV_r is (logarithmically) dependent on k_r . Thus, the main reason for the significant difference in EQE_{EL} ($\approx \frac{k_r}{k_{nr}}$), and thus ΔV_{nr} in the solar cells with different DA spacings is ascribed to the different k_{nr} . However, this does not suggest that k_r is completely independent of the DA distance.

In the **SI-6** of the revised manuscript, the discussion regarding the expected impact of the increased DA spacing on k_r is provided.

3) The transient voltage decay measurements should be explained in more detail (information in the SI is not enough). Why does the voltage decay time represent the lifetime of the photogenerated charges (i.e. the CT decay rate)? Does it not rather depend on extraction and free charge lifetime?

Our response: The voltage decay time does depend on the lifetime of free charge carriers, but extraction efficiency of charge carriers is irrelevant in the transient voltage decay measurements, because the measurements are done for the devices under an open-circuit condition: no charge carriers are extracted in the device under an open-circuit condition.

The relationship between the life time of charge carriers and the lifetime of CT states has been discussed in the literature (*Annu. Rev. Phys. Chem.* 2016. 67(113), 33), and more recently, in (*J. Phys. Chem. C* 2021, 125, 15590). Here, this relationship is simplified using the three-state model (**Figure R2**), in which the lifetime of charge carriers (τ_{FC}) is determined by the decay (k_{CT}) and the dissociation rate (k_{diss}) of CT state, as well as the decay rate of free charge carriers (k_{FC}), i.e., the rate of free charge carriers falling into CT states.

$$\tau_{FC} = \frac{1}{k_{CT}N_{CTC}} + N\left(\frac{1}{k_{diss}N_{CTC}} + \frac{1}{k_{FC}np}\right) \quad (\text{eq. R13})$$

where N represents the average number of times that the CT state would dissociate and

reform, before it decays to the ground state, N_{CTC} is the density of CT states, and n and p are the concentrations of free electrons and holes, respectively.

Figure R2. A three-state model describing the decay of free charge carriers (k_{FC}), the generation (G), dissociation (k_{diss}), and decay rate (k_{CT}) of CT states in a DA system.

Assuming that the generation rate of CT state is G , it can be derived (*Annu. Rev. Phys. Chem.* 2016. 67(113), 33):

$$N_{CTC} = \frac{G}{k_{CT}} \quad (\text{eq. R14})$$

$$k_{FC}np = \frac{k_{diss}G}{k_{CT}} \quad (\text{eq. R15})$$

Then, eq. R13 is reduced to

$$\tau_{FC} = \tau_{CT} + 2N\tau_{diss} \quad (\text{eq. R16})$$

where τ_{CT} is the lifetime of CT state, which equals to the inverse of the product of k_{CT} and N_{CTC} ; and τ_{diss} is the time needed for the CT state to dissociate into free charge carriers, which is the inverse of the product of k_{diss} and N_{CTC} . Therefore, the key determining factor for τ_{FC} (voltage decay lifetime determined by the TPV measurement) is τ_{CT} and τ_{diss} , and τ_{FC} linearly increases with τ_{CT} .

A more detailed discussion regarding the results from the transient voltage decay measurements are provided in **SI-7** of the revised manuscript.

4) There is not enough structural characterization (e.g. GIWAXS) for the BHJ blends, only the MD simulations are not convincing.

Our response: We now have measured GIWAXS for the BHJ blends, and the results are discussed in **SI-2** and **SI-3** of the revised manuscript.

Briefly, we observe from the TEM that the donor and the acceptor molecules in the active layers based on the BHJ systems are highly disordered. From the GIWAXS patterns of the pristine thin films of PBDB-T and PBDB-T(OD), lamellar structures with a vertical orientation are found. The equatorial scattering is at $q_x = 0.30 \text{ \AA}^{-1}$ for the film of PBDB-T, which is considerably larger than that of PBDB-T(OD) ($q_x = 0.25 \text{ \AA}^{-1}$), suggesting that the

distance between the adjacent layers in the lamella increases with the increasing size of the side chains. The GIWAXS patterns of the films of the BHJ systems based on different acceptors are very similar to that of the pristine donor polymers, showing lamellar structures. This suggests that the scattering signals mainly originate from the crystalline donor phases in the BHJ thin films. Also, we note that the distance between the layers in the lamella of the film of the BHJ system increases with the increasing size of the side chains. The GIWAXS results indicate that the side chains of the donor do play a crucial role in determining the arrangement of the molecules in the BHJ systems, in particular, the spacing between the molecules, agreeing well with the results from the MD simulations.

5) Apart from the D-A distance, the coupling also depends on the orientation of the molecules. How is this controlled in the studied systems?

Our response: We agree with reviewer that the coupling should depend on the orientation of the molecules in the DA system. According to the GIWAXS patterns shown in **SI-3**, all of the materials systems studied in this work have a face-on orientation.

REVIEWERS' COMMENTS

Reviewer #1 (Remarks to the Author):

The authors have made suitable revisions and addressed my concerns. I suggest the acceptance of the manuscript.

Reviewer #2 (Remarks to the Author):

I recommend publication in Nature Communications of this manuscript. However, I think that the authors should consider to address the following two points:

(1) My comment regarding the electro-phonon coupling was not addressed properly. The abstract still contain the statement: "Increasing DA spacing allows us to realize significantly weakened electron-phonon coupling, and thus, reduced k_{nr} and improved device voltage." There is no concrete discussion of the electron-phonon coupling in the paper except some unsupported statements.

According to Eq.1, electron-phonon coupling could impact both the electronic coupling matrix element (V) and Frank-Condon (FC) factor. So, electron-phonon coupling of which of these two parameters is affected by the DA distance? In the first sentence it is mentioned about " the inevitable electron-phonon coupling in organic materials" that I believe is due to the effect of high-frequency intramolecular modes on the FC factor. This effect, however, should marginally depend on DA distance.

(2) In their reply the authors provided a nice explanation how the DA distance impacts both radiative and nonradiative rates. The main text could benefit from a similar discussion. I suggest that, in addition to Eq.1, equation R5 is added and the text is revised following the same line as in the reply. The dependence of radiative rate also on $\Delta\mu$ that increases/decreases with an increase/decrease in DA distance is a key point here.

Reviewer #3 (Remarks to the Author):

The authors have very carefully responded to the comments of the three reviewers. They have added GIWAXS data, more explanations (e.g. effect of coupling on dV_r) and additional analysis (e.g. TMM). This has clearly improved the discussion and interpretations of the manuscript. Given the important message conveyed, the manuscript is now ready for publication in Nature Communications.

Thank you for considering our manuscript for publication in Nat. Commun. Below is our response to the reviewers' comments, and the changes made in the revised manuscript are marked in red.

Yours sincerely,
On behalf of all authors,
Weiwei Li, Koen Vandewal, Zheng Tang

Reviewer #1 (Remarks to the Author):

The authors have made suitable revisions and addressed my concerns. I suggest the acceptance of the manuscript.

Our response: We thank the reviewer for his/her positive feedback on our manuscript.

Reviewer #2 (Remarks to the Author):

I recommend publication in Nature Communications of this manuscript. However, I think that the authors should consider to address the following two points:

- (1) My comment regarding the electro-phonon coupling was not addressed properly. The abstract still contain the statement: "Increasing DA spacing allows us to realize significantly weakened electron-phonon coupling, and thus, reduced k_{nr} and improved device voltage." There is no concrete discussion of the electron-phonon coupling in the paper except some unsupported statements.

According to Eq.1, electron-phonon coupling could impact both the electronic coupling matrix element (V) and Frank-Condon (FC) factor. So, electron-phonon coupling of which of these two parameters is affected by the DA distance? In the first sentence it is mentioned about "the inevitable electron-phonon coupling in organic materials" that I believe is due to the effect of high-frequency intramolecular modes on the FC factor. This effect, however, should marginally depend on DA distance.

Our response: We thank the reviewer for his/her positive feedback and the suggestions to our manuscript. This statement "Increasing DA spacing allows us to realize significantly weakened electron-phonon coupling, and thus, reduced k_{nr} and improved device voltage" in the abstract is now deleted in the revised manuscript.

- (2) In their reply the authors provided a nice explanation how the DA distance impacts both radiative and nonradiative rates. The main text could benefit from a similar discussion. I suggest that, in addition to Eq.1, equation R5 is added and the text is revised following the same line as in the reply. The dependence of radiative rate also on $\Delta\mu$ that increases/decreases with an increase/decrease in DA distance is a key point here.

Our response: In the revised manuscript, we have added a sentence describing the determining factors for k_r , and used a reference to support our argument regarding the possibility to reduce

knr without increasing kr. Also, we have added the discussion about the impact of DA spacing on kr, and explained the reason for the similar radiative voltage losses in the solar cells with different DA spacings. The changes made in the revised manuscript are marked in red.

Reviewer #3 (Remarks to the Author):

The authors have very carefully responded to the comments of the three reviewers. They have added GIWAXS data, more explanations (e.g. effect of coupling on dVr) and additional analysis (e.g. TMM). This has clearly improved the discussion and interpretations of the manuscript. Given the important message conveyed, the manuscript is now ready for publication in Nature Communications.

Our response: We thank the reviewer for his/her positive feedback on our manuscript.